# Within-Subtype HIV-1 Polymorphisms and Their Impacts on Intact Proviral DNA Assay (IPDA) for Viral Reservoir Quantification

**DOI:** 10.3390/v17111453

**Published:** 2025-10-31

**Authors:** Mohith Reddy Arikatla, Jyoti S. Mathad, Kavidha Reddy, Nicole Reddy, Thumbi Ndung’u, Kathryn M. Dupnik, Guinevere Q. Lee

**Affiliations:** 1Infectious Diseases Division, Department of Medicine, Weill Cornell Medical College, New York, NY 10065, USA; moa4020@med.cornell.edu (M.R.A.); kad9040@med.cornell.edu (K.M.D.); 2Center for Global Health, Weill Cornell Medicine, New York, NY 10065, USA; 3Africa Health Research Institute, Durban 4001, South Africa; 4Division of Infection and Immunity, University College London (UCL), London WC1E 6JF, UK; 5HIV Pathogenesis Programme, The Doris Duke Medical Research Institute, University of KwaZulu-Natal, Durban 4001, South Africa; 6Ragon Institute of Massachusetts General Hospital, Massachusetts Institute of Technology, and Harvard University, Cambridge, MA 02139, USA; 7Department of Microbiology and Immunology, Weill Cornell Medical College, New York, NY 10065, USA

**Keywords:** HIV proviruses, intact proviruses, HIV diversity, IPDA, intra-subtype variation, phylogenetics, PCR, India, subtype C

## Abstract

The Intact Proviral DNA Assay (IPDA) is widely used to quantify genome-intact HIV proviruses in people living with HIV, but viral sequence diversity has been observed to cause assay failures due to primer/probe mismatches. Adapted for subtype C, IPDA-BC is a modified version of the IPDA validated on South African HIV-1 subtype C. India is also impacted by subtype C, but IPDA performance within-subtype across geographical regions is not well studied. We analyzed Indian (IN) and South African (ZA) subtype C sequences in silico, hypothesizing that IPDA-BC may underperform with IN viruses. Primer/probe binding was predicted using three increasingly stringent nucleotide mismatch criteria, whose sensitivity and specificity were evaluated against experimental IPDA outcomes. Phylogenetic analyses confirmed that IN and ZA subtype C sequences form distinct clusters with significant compartmentalization (*p* < 0.003). Across criteria, up to 6–10% decreases in primer/probe binding were observed in IN versus ZA, with the *env* forward primer being the most affected. These criteria showed low sensitivity (18–53%) and variable specificity (67–100%) in predicting experimental outcomes. In conclusion, even within subtype, HIV-1 variation across geographical regions may impact IPDA performance, underscoring the need for improved predictive models to guide assay design for global HIV cure research.

## 1. Introduction

The Intact Proviral DNA Assay (IPDA) [1] is a highly sensitive PCR-based molecular technique used to quantify the number of intact HIV proviruses in the cells of people living with HIV. This assay is crucial for HIV cure research because it distinguishes between genetically intact proviruses, which could potentially reactivate and produce infectious virus, and defective proviruses, which lack the ability to replicate [2]. Though the majority of integrated HIV is defective or incomplete, the intact proviruses pose a significant barrier to achieving a cure as they result in viral rebound if antiretroviral therapy (ART) is stopped.

Prior to IPDA, molecular methods like total HIV DNA quantification via quantitative PCR (qPCR) or single-target Droplet Digital PCR (ddPCR) were unable to differentiate between defective and intact proviruses [3], making it difficult to measure the true size of the replication-competent reservoir. Viral outgrowth assays and near-full-genome viral DNA sequencing more accurately assess provirus intactness [3], but these assays are labor-intensive and expensive. Additionally, viral outgrowth assays require a large number of cells, which precludes their use outside of specialized research centers. In contrast, IPDA provides a relatively precise and economical quantification of intact proviral genomes. IPDA has become instrumental for assessing the effectiveness of potential cure strategies aimed at reducing or eliminating the intact reservoirs, especially in more resource-limited settings. As a result, IPDA plays a key role in evaluating the success of potential HIV cure interventions, such as latency-reversing agents and gene therapies, by allowing researchers to monitor changes in the size of the intact reservoir over time in a high-throughput manner.

The original IPDA, hereafter referred to as IPDA-original, was developed based on subtype B HIV-1 sequences [1]. However, HIV-1 is genetically diverse, and even for subtype-B sequences, the IPDA-original primers and probes have a high failure rate due to viral polymorphisms: 28% in one study [4] and 18% in another [5]. Furthermore, multiple studies have shown that non-B HIV-1 subtypes, which make up over 90% of the HIV epidemic worldwide [6], have viral sequence polymorphisms that could lead to primer/probe mismatches and subsequent IPDA failures [4,7,8]. To address this issue, multiple research groups have adapted the IPDA-original for use in other subtypes. For example, a cross-subtype IPDA (CS-IPDA) was developed for subtypes A, B, C, D, and CRF01_AE [9]; a modified version of IPDA called IPDA-A1D was developed for subtype A1 and D [8]; and another modified version of IPDA was developed for subtypes B and C HIV-1 [10], hereafter referred to as IPDA-BC, which we use as an example for evaluation in this study.

IPDA-BC was designed based on 148 reference genomes from the 2018 Los Alamos HIV Sequence Compendium [10]. The primers and probes were bioinformatically validated with 2125 (randomly subset to 752) subtype B sequences derived from the USA SCOPE and OPTION cohorts obtained via the FLIPS near-full-length proviral genome sequencing approach [11], and 697 subtype C HIV-1 sequences derived from 24 South African FRESH (Females Rising through Education, Support, and Health) cohort participants obtained via the FLIP-seq near-full-length proviral genome sequencing approach [12,13]. IPDA-BC was further experimentally validated with Gblocks and five South African subtype C HIV-1 samples from the same FRESH cohort study participants. Unlike IPDA-original, neither the IPDA-BC *psi* probe nor its primers bind to the viral Major Splice Donor Site (MSD, HXB2 744), although the resulting *psi* amplicon does include the MSD site. Compared to IPDA-original, IPDA-BC shifts all *psi* primer and probe targets to more conserved genomic regions shared by both subtypes B and C HIV-1. For *env* forward, *env* reverse, and *env*-hypermutated primers/probes, IPDA-BC retained IPDA-original’s targeted genomic sites. The IPDA-BC *env*-intact probe introduces two additional bases at the 3’ end of IPDA-original’s *env*-intact probe, thus making it the same length as the *env*-hypermutated probe. A detailed comparison of the primers and probes from IPDA-original and IPDA-BC is shown in Appendix A.

As mentioned above, IPDA-BC was primarily validated using South African subtype C HIV-1 sequences. However, India is also impacted predominantly by subtype C HIV-1 [14]. Due to the polymorphic nature of HIV and intra-subtype sequence differences across geographical regions, we hypothesize that Indian subtype C HIV-1 will be genetically distinct from South African subtype C HIV-1, and that IPDA-BC will exhibit a higher in silico failure rate for subtype C HIV-1 sequences isolated in India than those in South Africa. If confirmed, this would suggest that assays developed and validated on viral sequence diversity from a specific region should be adapted cautiously when applied elsewhere. This, to our knowledge, is the first study to evaluate whether identical viral subtypes circulating in geographically distinct regions would require further IPDA adaptations.

## 2. Methods

### 2.1. Phylogenetic Analysis

For the phylogenetic comparison of subtype C HIV-1 isolated in India (IN) and South Africa (ZA), we downloaded all available subtype C sequences in these regions from the Los Alamos HIV Sequence Database (available online at https://www.hiv.lanl.gov/content/sequence/HIV/mainpage.html (accessed on 18 June 2025)) [14] spanning HXB2 coordinates 400-1400 (covering IPDA-*psi*) and 7000–8000 (covering IPDA-*env*) (Figure 1). Sequence sets from each genomic region were subjected to multiple sequence alignment using Muscle (version 3.50.0, available online at https://bioconductor.org/packages/release/bioc/html/muscle.html (accessed on 18 June 2025)) [15] followed by neighbor-joining tree construction using the R package ape (version 5.8.1, available online at https://cran.r-project.org/web/packages/ape/index.html (accessed on 18 June 2025)) [16]. Formal statistical tests for compartmentalization were performed using the Slatkin-Maddison compartmentalization test implemented in the HyPhy package (version 2.5.74, available online at https://github.com/veg/hyphy (accessed on 18 June 2025)) [17].

### 2.2. In Silico Evaluation of the Compatibility of IPDA Primers/Probes per Geographical Region

All available subtype C HIV-1 sequences labeled with the country codes IN (India) and ZA (South Africa), and subtype B HIV-1 sequences labeled with country code US (United States) were downloaded from the Los Alamos HIV Database Search Interface [14] on 18 June 2025. Each of these sequences were queried against the HXB2 reference genome using NCBI BLAST+ suite (version 2.15.0, available online at https://www.ncbi.nlm.nih.gov/books/NBK131777/ (accessed on 18 June 2025)) [18] and interrogated for the presence of IPDA-original and IPDA-BC primer/probe binding sequences (Appendix A) using an in-house R algorithm (available online at https://github.com/guineverelee/HIVprimertestR (accessed on 18 June 2025)). We evaluated three different definitions to predict primer/probe binding failures due to sequence mismatches (Figure 2a). In our first definition (Definition #1), we used the same criteria for primer/probe mismatch as those published by Gaebler et al. [19], which were also used in the development of IPDA-BC [10]. In this definition, primer target regions were split evenly into two halves: a 5’ end and a 3’ end. If the primer target region had an odd number of nucleotides, the middle nucleotide was allocated to the 5’ end. Up to three single-nucleotide mismatches were allowed within the 5’ end, whereas a maximum of one mismatch was allowed in the 3’ end. Only one single-nucleotide mismatch was allowed for the probe region. In our second definition (Definition #2), we increased stringency by failing any sequences with mismatches in the last two bases of the 3’ end of the forward or reverse primers. This is because a lack of nucleotide mismatches at the 3’ end of a primer target region had been shown to be crucial for successful PCR amplification [20]. All other criteria remained the same as Definition #1. In our last definition (Definition #3), only sequences with 100% sequence identity to the respective primer/probe were considered a pass. Definition #3 represents the minimum probable fraction of sequences that would successfully yield IPDA signals. All analyses were performed in R. All R scripts are publicly available (see GitHub link above). This script package is not restricted to IPDA evaluation; it can also be applied to any future HIV-1 primer/probe designs against any background population sequence datasets according to Definitions #1–3.

## 3. Results

### 3.1. Subtype C HIV-1 Isolated from India (IN) and South Africa (ZA) Were Genetically Distinct

From the Los Alamos HIV Sequence Database [14], we retrieved 10 IN and 29 ZA subtype C sequences spanning HXB2 coordinates 400–1400 (covering IPDA-*psi*), and 746 IN and 8990 ZA subtype C sequences spanning HXB2 7000–8000 (covering IPDA-*env*). Phylogenetic analyses showed that the IN (subtype C) and ZA (subtype C) isolates separated into their own respective clusters (Figure 1 and Appendix A). Significant geographical compartmentalization was confirmed by Slatkin-Maddison compartmentalization tests (*psi p* < 0.003 and *env p* < 0.001). These results suggest that a subtype-specific IPDA validated on South African subtype C HIV-1 sequences should be evaluated before being applied to Indian subtype C HIV-1 samples.

### 3.2. In Silico Analysis Predicted That IPDA-BC Primers/Probes Were 6–10% More Likely to Fail in IN than ZA Samples

As mentioned in the Section 2, we used three different definitions to predict whether a primer/probe would bind (Figure 2a). Definition #1 was previously published and has been used in multiple IPDA-related development studies [10,19]. Referring to Figure 2b,c, the IPDA-original design was predicted to perform poorly in the *psi* probe region especially in IN (C) and ZA (C) sequences, whereas the IPDA-BC *psi* probe design would rescue almost all sequences from IN (C), ZA (C), and US (B). Using this definition, IPDA-BC would result in a maximum of 6% increase in binding failures if used for IN (C) samples (*env* forward primer had the maximum reduction from 96% ZA (C) to 90% IN (C)). Definition #2 is a more stringent version of Definition #1 with an added requirement that the last two bases at the 3’ of a primer target site must be identical to the primer sequence. As shown in Figure 2d,e, IPDA-BC remained more appropriate for samples across the three geographical regions relative to IPDA-original. Using this definition, IPDA-BC would result in a maximum increase in binding failures of 6% if used for IN (C) samples (*env* forward primer had the maximum reduction from 95% ZA (C) to 89% IN(C)). Definition #3 is the most stringent and requires 100% sequence identity against each primer or probe. As shown in Figure 2f,g, ≤1% of IN (C) and ZA (C) sequences had complete sequence identity against the IPDA-original design at the *psi* probe and *env* forward primer target sites, whereas the IPDA-BC design significantly improved the inferred success rate. Using this definition, IPDA-BC would result in a maximum of 10% more binding failures if used for IN (C) samples (*env* forward primer had the maximum reduction from 63% to 53%). The use of Definition #3 also revealed the minimum probable fraction of sequences within a population that were associated with binding successes. At minimum, IPDA-BC primers/probes exhibited 100% sequence identity with only 53% of IN (C) (*env* forward), 63% of ZA (C) (*env* forward), and 64% of US (B) (*env* forward) sequences in this analysis. The exact type of target sequence versus primer/probe mismatches are summarized in Appendix A.

The analysis shown in Figure 2 was not adjusted for donor-specific over-representation due to multiple sequences derived from the same donor within the query dataset, and query sequences were not screened for mapping abnormalities such as presence of large gaps. Therefore, we then repeated the analyses with a quality filter on the target sequences to remove mappings that were either <50% or >150% of the primer/probe sequence length and sequences that exhibit <50% sequence identity with the respective primer/probe (Appendix A). In addition to the quality filter, we also generated a donor-wise majority consensus sequence in cases where donor information was available (Appendix A). Both additional analyses led to similar conclusions as the first analysis: IPDA-BC showed a significant improvement from IPDA-original for both IN (C) and ZA (C) samples. With the sequence quality filter alone, IN (C) samples were predicted to have a 6–10% reduction in capture relative to ZA (C) samples, whereas applying the sequence quality filter with donor-wise consensus predicted a 2–9% reduction. All plotting data and the number of query sequences analyzed for each geographic region are available in Appendix A.

Since IPDA-BC was designed to incorporate both subtype-B and subtype-C-specific primers into the same reaction (Appendix A), all the above analyses were performed with the assumption that both B and C primers were present in the ddPCR reaction setup. We performed an additional sub-analysis on a subtype-matched ddPCR reaction setup scenario. Our analyses revealed that a subtype-matched approach further reduced binding success rate in all US (B), IN (C), and ZA (C) (Appendix A). For example, according to Definition #3, not using the IPDA-BC *psi* reverse-B primer for IN (C) and ZA (C) reduced the capture by 11% and 35%, respectively. Similarly, not using IPDA-BC *psi* reverse-C for US (B) reduced the minimum predicted capture fraction by 15%. All plotting data and the number of query sequences associated with this set of analyses are available in Appendix A.

### 3.3. Ability to Distinguish Between Hypermutated and Non-Hypermutated Genomes

Next, we evaluated whether the IPDA-original and IPDA-BC *env*-hypermutated probe designs were likely able to distinguish between APOBEC-3G/3F-mediated hypermutated proviral genomes versus non-hypermutated ones. By design [1,10], the *env*-hypermutated probe serves as a competing non-fluorescent probe with nucleotides “A and A” in the designated hypermutation sites as opposed to having “G and G” nucleotides in the *env*-intact fluorescent probe (Appendix A, red fonts). For each geographical region, we quantified the number of query sequences with ≤1 single-nucleotide mismatch that also naturally contain “G and A” or “A and G” in these two positions and would qualify for successful probe binding with both *env*-hypermutated and *env*-intact probes according to Definitions #1 and #2. For both Definitions #1 and #2, approximately 2% of IN (C), 8% of ZA (C), and 3% of US (B) sequences had a “G and A” or “A and G” genotypes at these positions, representing the fraction of sequences in these populations for which the *env*-hypermutated probe would unlikely function as an effective competitor to distinguish *env*-hypermutated genomes. Since Definition #3 requires zero mismatches, no queries with “G and A” or “A and G” at these positions would qualify for either of the *env* probes.

### 3.4. Performance Evaluation of the Three Mismatch Definitions Used in This Study

In this study, we based our evaluation on whether a primer or probe would successfully bind to a target sequence using a previously employed criteria for IPDA-BC development (Definition #1), and we further increased the stringency of the passing criteria in Definitions #2 and #3. However, PCR success for HIV amplification is known to depend on additional factors such as the nature of the nucleotide mismatch and its relative position in the primer/probe, as exemplified by the study by Kinloch et al. [4] where IPDA failures were not associated with any obvious mismatch patterns that would collectively predict IPDA failures. In this context, we evaluated the sensitivity and specificity of Definitions #1, #2, and #3 in predicting experimental IPDA successes and failures using 23 samples with matched IPDA results and viral sequence data previously published by the eCLEAR clinical trial [21] (Appendix A). Definitions #1, #2, and #3 showed 53%, 47%, and 18% sensitivity in predicting IPDA experimental success, and 67%, 67%, and 100% specificity, respectively. In a separate experiment, IPDA-BC was applied to South African subtype C HIV-1 samples derived from five FRESH cohort donors (Appendix A) and resulted in no failures as expected. When applied to donor-matched consensus sequences [13], Definitions #1, #2, and #3 showed 60%, 60%, and 0% sensitivity in predicting IPDA-BC experimental success for these samples. Specificity could not be evaluated because none of the samples failed IPDA-BC.

## 4. Discussion

HIV-1 is genetically diverse. Most PCR-based molecular biology assays, sequencing techniques, and bioinformatics methods for viral reservoir characterization in clinical samples developed based on our understanding of the sequence diversity in one HIV-1 subtype (often subtype B) must be adapted for application to non-B HIV-1 subtypes. Importantly, even within a given subtype, viral sequences exhibit significant diversity. In this study, we used subtype C HIV-1 isolated in India versus South Africa as an example to conduct an in silico evaluation of whether an assay such as IPDA-BC, which was validated largely based on South African subtype C HIV-1 strains, would require further assay adaptations before its application to subtype C strains in India. Based on our evaluation, we concluded that the possibility of a signal failure for IPDA-BC would likely marginally increase for Indian subtype C samples. However, our conclusion is limited by the relatively poor predictive values of these three criteria.

Despite the poor predictive values associated with our Definitions #1–3, we nevertheless showed that in a curated, centralized HIV genomics database, Indian subtype C HIV-1 sequences were genetically distinct from South African subtype C HIV-1 sequences, showing an up to 6% decrease in predicted primer/probe capture according to Definitions #1 and #2 and an up to 10% decrease according to Definition #3. Our results suggest that for Indian HIV-1 reservoir studies, instead of adopting the IPDA-BC approach as-is, researchers may need to first survey viral sequence diversity within a cohort of interest by sequencing the virus, then further adapt the primers/probes for the viral polymorphisms found in the respective geographic region. These will need to be periodically re-evaluated because of shifting epidemiology and the emergence of circulating recombinant strains. Globally, circulating recombinant strains account for about a third of HIV infections, followed in prevalence by subtype C at 23% [6]. Subtype C predominates in eastern and southern Africa and is about 90% of the HIV-1 circulating in India [22]. Subtype C can emerge as the dominant strain at regional levels as well, as has been seen in southern Brazil [23]. HIV-1 sequence diversity will likely be a continuous challenge when adapting IPDA and similar PCR-based reservoir quantification assays to different HIV-1 subtypes globally. We expect the same challenges will be faced by similar assays such as ddPCR-based HIV-1 RNA transcript profiling assays [24,25].

Our observations also highlight the importance of using IPDA-BC with both subtype B and C primers in the ddPCR reaction as-is: We showed that modifying IPDA-BC by removing the subtype B primers in the ddPCR reaction mix when applying the assay to subtype C samples reduced the predicted capture success. Likewise, modifying IPDA-BC by removing the subtype C primers in the ddPCR reaction mix when applying the assay to subtype B samples also reduced the predicted capture success.

Based on our findings in this study, the higher rate of sequence mismatches with primers/probes could result in an underestimation of total and intact proviral load. Consequently, there may be a higher proportion of negative results (assay failure) in clinical subtype C samples obtained from India compared to those from South Africa. However, the magnitude of underestimation is likely to vary across donors and geographic regions. For example, even within the same region, proviral sequence diversity is expected to be higher in individuals who experienced prolonged viremia than in those who initiated ART during acute infection (e.g., FRESH cohort [13,26] and the cohort in [27]). This is further complicated by the degree of nucleotide mismatches between the viral strain within each individual and the IPDA-BC primers/probes. Our findings may also have implications in IPDA-derived intact/defective proviral genome ratio values. For instance, if more intact viruses contain *env* regions with primer/probe mismatches in Indian samples relative to South African samples but this bias is absent in defective genomes, there would be an inflation of the defective/intact proviral genome ratio in Indian samples. To address this concern, large-scale full-length proviral sequencing of Indian samples is required; however, such a database does not currently exist. We therefore urge future studies to generate such data. These considerations reflect the inherent nature of the IPDA design: the presence of double-positive *psi* and *env* signals represents a high likelihood of genome-intact proviruses, whereas the absence of a signal may indicate either true absence or it can be an indicator of primer/probe mismatches [28]. In this context, the assay is best suited for longitudinal monitoring of changes in intact reservoir size within an individual, as reservoir diversity is expected to remain relatively comparable across time points within the same donor on suppressive therapy. Accordingly, IPDA double-positive signals should be interpreted as the minimal estimate of intact proviral load.

Our study is foremost limited by the poor sensitivity and specificity of the prediction criteria we used. As expected, all cases of eCLEAR trial experimental IPDA failures occurred when a target sequence had less than 100% sequence identity relative to the primers/probes (Appendix A), supporting that Definition #3 represents the probable minimal passing frequency. Despite the absence of failures in the FRESH cohort (the original samples used to validate IPDA-BC), our in silico Definition #1 prediction achieved only 60% sensitivity. We also recognize that our approach to evaluate the sensitivity and specificity of the in silico Definitions used to predict primer/probe binding may be influenced by the use of viral sequences obtained from pre-ART plasma samples of eCLEAR cohort donors. Specifically, sequences from pre-ART plasma may represent only a subset of the proviral pool during supressive ART [29]. Futhermore, we used consensus sequences derived from donor-specific viral pools for the sensitivity and specificity evaluations. Since consensus sequences represent the most frequently observed bases at each nucleotide position and may miss minority variants, our sensitivty and specificity estimation should be considered a minimal estimate.

Despite these caveats, our observations revealed the limitations of simple in silico prediction algorithms for PCR success, especially when the target is as genetically diverse as HIV. In this study, we attempted to predict IPDA experimental success based on nucleotide mismatch count, but PCR success is also influenced by a range of other factors such as annealing temperature and ionic concentration [30]. Since some current assays were designed based on Definition #1, our findings highlight the urgent need for the development of better predictive algorithms. Further studies should be conducted using large-scale datasets with viral sequences from diverse HIV-1 subtypes with matching IPDA pass/fail data, followed by machine learning strategies to develop more accurate in silico prediction algorithms.

In addition to the limited sensitivity and specificity values as discussed above, our conclusions are further limited by these factors: First, our findings are limited by the low number of subtype C Indian viral sequence data available through the Los Alamos HIV Sequence database [14], especially at the *psi* region (Appendix A). The minimum number of Indian *psi* sequences analyzed was 73, 73, and 34 in Analyses 1, 2, and 3, respectively, underscoring substantial gaps in our understanding of HIV sequence polymorphisms in India. Second, the dataset available in the Los Alamos HIV Sequence Database may not represent contemporary viral strains circulating in India. Third, we were unable to evaluate the sensitivity and specificity of IPDA-BC on subtype C samples from India due to the lack of IPDA data with paired near full length proviral sequences data from which genome-intactness can be determined. This limitation prevented us from determining whether IPDA-BC introduces systematic bias or misclassification of intact and defective genomes in Indian samples. To address these gaps, future efforts should prioritize expanding sequencing coverage of underrepresented genomic regions from subtype C HIV in India, as well as generating paired IPDA and proviral sequence data.

In conclusion, primer and probe design will remain a persistent challenge in PCR-based assay development due to the extensive sequence diversity of HIV-1. Although both India and South Africa are affected by subtype C HIV-1, based on the data available in the Los Alamos HIV Sequence Database [14], our findings demonstrate that the circulating strains in these regions are genetically distinct. Assays such as IPDA-BC should thus be applied with caution and appropriate customizations when used in genetically diverse populations.

## Figures and Tables

**Figure 1 viruses-17-01453-f001:**
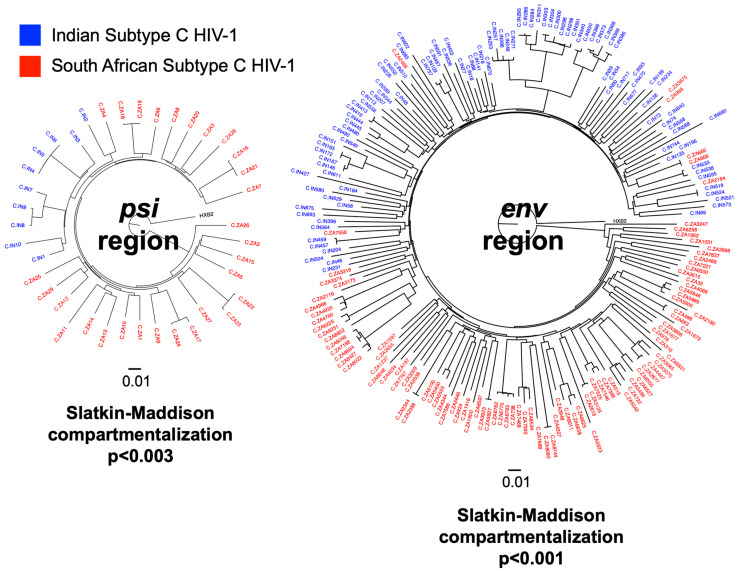
Indian and South African Subtype C HIV-1 are phylogenetically distinct. HIV-1 subtype C sequences associated with India (IN) and South Africa (ZA) downloaded from the Los Alamos HIV Sequence Database [14] spanning HXB2 coordinates 400-1400 (covering IPDA-*psi*; 10 IN and 29 ZA sequences) and 7000-8000 (covering IPDA-*env*; 746 IN and 8990 ZA sequences). All *psi* region sequences are represented in the phylogenetic tree. For *env*, due to the large number of sequences available, the sequence sets from each country were bootstrapped 10 times at 100 sequences per set. A representative *env* tree is shown here (see Appendix A for all bootstrapped *env* trees). The sizes of the polar phylogenetic trees represent relative genetic diversity. Both trees are rooted to HXB2. IN and ZA sequences are significantly compartmentalized by Slatkin-Maddison tests.

**Figure 2 viruses-17-01453-f002:**
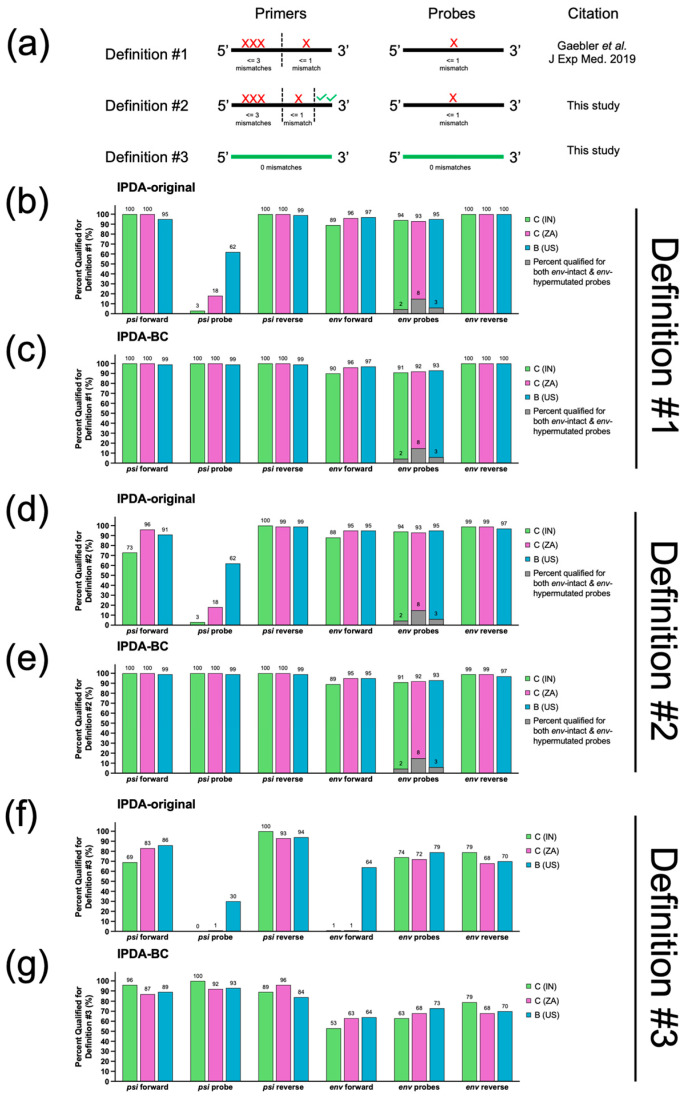
In silico successful binding rate of IPDA-original and IPDA-BC primers/probes against Indian (IN) and South African (ZA) subtype C HIV-1 sequences. (**a**) Three in silico definitions of binding success were examined in this study. Criteria used for Definition #1 are the same as those published by Gaebler et al. [19]. (**b**–**g**) Per definition of success, we evaluated the fraction of Indian (IN), South African (ZA) and USA (US) HIV-1 sequences that are predicted to bind to each of the primers and probes of both IPDA-original and IPDA-BC. Detail analysis tables are available in Appendix A.

## Data Availability

The sequences representing HIV-1 subtype C Indian (IN) and South African (ZA), and subtype B USA (US) were downloaded from the Los Alamos HIV sequence database [https://www.hiv.lanl.gov (accessed on 18 June 2025)] [14]. All plotting data and analyses presented in this study are included in the article/Appendix A. Further inquiries can be directed to the corresponding author. The experimental IPDA-BC results and the donor-wise consensus sequences used to evaluate the sensitivity and specificity of Definitions #1–3 were obtained from the Appendix A of the eCLEAR trial publication [21]. Viral sequence data associated with the FRESH cohort were previously published in Reddy et al. [13].

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
