# Peer review of "Within-Subtype HIV-1 Polymorphisms and Their Impacts on Intact Proviral DNA Assay (IPDA) for Viral Reservoir Quantification"

_viruses, 2025, doi:10.3390/v17111453_

Round 1
Reviewer 1 Report
Comments and Suggestions for Authors
In the manuscript by Arikatla and colleagues, the authors evaluate HIV-1 polymorphisms using a redesigned "Intact Proviral DNA Assay" (IPDA) and their impact on viral reservoir quantification. Additionally, the authors also state that, "Though the majority of integrated HIV is defective or incomplete, the intact proviruses pose a significant barrier to achieving a cure as they result in viral rebound if antiretroviral therapy (ART) is stopped," which is also a correct statement. Here, the authors hypothesize that such assays are essential for monitoring reactivation and production of intact virus versus defective proviruses. This statement is not correct since defective proviruses also integrate into the host genome and can be reactivated (assuming mutations are not within the genes (structural proteins) required for the formation of virus particles. Even sequencing the DNA from integrated proviruses does not guarantee that they can be reactivated. Thus, the only methods that will determine whether the virus particles are infectious (and pointed out by the authors) are the virus outgrowth assay (VOA) or molecular cloning of the viral RNA and virus rescue. Since the authors are using only two oligonucleotide primer sets complementary to the psi site and the env gene (covering roughly 2000 bases of a genome that is greater than 9000 bases), they will not be able to determine if the integrated virus is viable. Hence, the results will equivocal.
Major comments:
1) Figure 1 is unreadable.
2) Supplementary Figure 2 is impossible to interpret.
Minor comments:
Line 55: In the sentence, “effectiveness of cure strategies aimed at” should be ” effectiveness of potential cure strategies aimed at.”
Line 57: In the sentence with, “the success of various HIV cure” should be changed to “ the success of potential HIV cures.”
Line 65: Reference 6 should be moved to the end of the sentence.
Lines 87-89: Does adding two nucleotides result in a frame shift?
Line 145: In the sentence with “sequence identity against the respective,” would sound better as “sequence identity to the respective.”
Comments on the Quality of English LanguageThere were just a few errors, but overall it's fine.
Author Response
Comment 1: In the manuscript by Arikatla and colleagues, the authors evaluate HIV-1 polymorphisms using a redesigned "Intact Proviral DNA Assay" (IPDA) and their impact on viral reservoir quantification. Additionally, the authors also state that, "Though the majority of integrated HIV is defective or incomplete, the intact proviruses pose a significant barrier to achieving a cure as they result in viral rebound if antiretroviral therapy (ART) is stopped," which is also a correct statement. Here, the authors hypothesize that such assays are essential for monitoring reactivation and production of intact virus versus defective proviruses. This statement is not correct since defective proviruses also integrate into the host genome and can be reactivated (assuming mutations are not within the genes (structural proteins) required for the formation of virus particles. Even sequencing the DNA from integrated proviruses does not guarantee that they can be reactivated. Thus, the only methods that will determine whether the virus particles are infectious (and pointed out by the authors) are the virus outgrowth assay (VOA) or molecular cloning of the viral RNA and virus rescue. Since the authors are using only two oligonucleotide primer sets complementary to the psi site and the env gene (covering roughly 2000 bases of a genome that is greater than 9000 bases), they will not be able to determine if the integrated virus is viable. Hence, the results will equivocal.
Response 1: We appreciate the reviewer pointing out a potential source of misinterpretation in our text: “This assay is crucial for HIV cure research because it distinguishes intact proviruses, which are potentially capable of reactivating and producing infectious virus, from defective proviruses, which cannot replicate[2].” In lines 41-43.
Our intent here was to define HIV-1 genome intactness while explaining the function of IPDA. To reduce ambiguity, we have revised the sentence to “This assay is crucial for HIV cure research because it distinguishes between genetically intact proviruses, which could potentially reactivate and produce infectious virus, and defective proviruses, which lack the ability to replicate[2].” (lines 41-43).
Major comments:
1) Figure 1 is unreadable.
2) Supplementary Figure 2 is impossible to interpret.
Response to major comments:
We thank you for pointing out the quality of the figures and apologize for our oversight. We have requested the editors to replace all the figures with higher resolution images.
Minor comments:
Line 55: In the sentence, “effectiveness of cure strategies aimed at” should be ” effectiveness of potential cure strategies aimed at.”
Line 57: In the sentence with, “the success of various HIV cure” should be changed to “ the success of potential HIV cures.”
Line 65: Reference 6 should be moved to the end of the sentence.
Lines 87-89: Does adding two nucleotides result in a frame shift?
Line 145: In the sentence with “sequence identity against the respective,” would sound better as “sequence identity to the respective.”
Response to minor comments:
We added “potential” to lines 55 and 57 as suggested.
In line 66 (previously 65), we would prefer to retain the current placement of reference 6, as it specifically supports the statement that non-B HIV-1 subtypes constitute over 90% of the global HIV epidemic. Moving it to the end of the sentence could create ambiguity regarding which part of the statement the citation substantiates.
In lines 88-90 (previously 87-89), we described the probe design of IPDA-BC, which differentiates between env-intact and env-hypermutated proviruses. A two-nucleotide addition in the probe will not result in frameshift in the template. To ensure clarity, we added lines 90-91: “A detailed comparison of the primers and probes from the original IPDA and IPDA-BC is shown in Supplementary Figure 1.”, which provides a side-by-side comparison of the probe sequences used in both assays.
We revised the text in line 146 (previously 145) to “sequence identity to the respective primer/probe”.
Comments on the Quality of English Language
There were just a few errors, but overall, it's fine.
Response: We thank the reviewer for the feedback and have made minor changes through the manuscript.
Reviewer 2 Report
Comments and Suggestions for Authors
Monitoring HIV latent reservoirs is a very important and the manuscript: "Within-subtype HIV-1 polymorphisms and their impacts on Intact Proviral DNA Assay (IPDA) for viral reservoir quantification" is highlighting a frequent issue with HIV RNA/DNA amplification; primer mismatch. The introduction is informative enough and materials and methods adequately describe the authors' research and criteria for the results. Conclusion is supported by the results, and the result presentation, supplementary files included is very extensive. While I find this paper may be published in the present form, I would advise the authors to expand the conclusion section with some limitations of the study, such as limited number of the sequences included.
Author Response
Comments and Suggestions for Authors
Monitoring HIV latent reservoirs is a very important and the manuscript: "Within-subtype HIV-1 polymorphisms and their impacts on Intact Proviral DNA Assay (IPDA) for viral reservoir quantification" is highlighting a frequent issue with HIV RNA/DNA amplification; primer mismatch. The introduction is informative enough and materials and methods adequately describe the authors' research and criteria for the results. Conclusion is supported by the results, and the result presentation, supplementary files included is very extensive. While I find this paper may be published in the present form, I would advise the authors to expand the conclusion section with some limitations of the study, such as limited number of the sequences included.
Response: We sincerely thank the reviewer for the positive feedback. As suggested, we have expanded on the limitations of our study in lines 322-336 and 347-361 of the Discussion. Specifically, we now have elaborated on the limited sensitivity and specificity of the in silico prediction criteria used in this study, and how the evaluation of these criteria may be influenced by the nature of the sequences analyzed. We have also addressed the limited number of sequences representing subtype C HIV from India and discuss the potential biases introduced by such small datasets. Finally, we acknowledge the lack of experimental validation of IPDA-BC on Indian subtype C samples in our study and urge future studies to capture near full length proviral genomes from this geographical region. To our knowledge, there are currently no studies that have evaluated near-full-length proviral genome sequences from Indian samples.
Reviewer 3 Report
Comments and Suggestions for Authors
The paper by Arikatla et al explores the important general question as to how to reliably measure the HIV reservoir. They focus on the intact proviral DNA assay (IDPA), which has been used to gain insights into persistent intact and defective HIV genomes in PWH. In particular, the investigators address the impact of sequence variation in viruses of the same clade that were obtained from different geographical regions. They utilize previously identified viruses from South Africa and India for primer binding sites that are typically targeted for IPDAs and then determine in silico what are potential failure rates. Overall, they observe a great deal of variation between the viruses from S. Africa and India. They also model that this variation would be predicted to lead to greater IPDA failure rates. Implications are that primers should be optimized, not only for clades, but different cohorts.
My major concern is how does this actually translate to clinical samples and how would the failure rates bias data from biological samples. For example, would there potentially be more negative samples or an overrepresentation of defective viruses. Does this actually skew the data from clinical samples in which the IPDA will not be informative such as relative differences in proviral genomes? How much does this variation actually underestimate the "number" of cells harboring HIV? Some of their findings would be potentially addressed if they performed IPDA utilizing samples from South Africa and India.
Author Response
Comment 1:
The paper by Arikatla et al explores the important general question as to how to reliably measure the HIV reservoir. They focus on the intact proviral DNA assay (IDPA), which has been used to gain insights into persistent intact and defective HIV genomes in PWH. In particular, the investigators address the impact of sequence variation in viruses of the same clade that were obtained from different geographical regions. They utilize previously identified viruses from South Africa and India for primer binding sites that are typically targeted for IPDAs and then determine in silico what are potential failure rates. Overall, they observe a great deal of variation between the viruses from S. Africa and India. They also model that this variation would be predicted to lead to greater IPDA failure rates. Implications are that primers should be optimized, not only for clades, but different cohorts.
My major concern is how does this actually translate to clinical samples and how would the failure rates bias data from biological samples. For example, would there potentially be more negative samples or an overrepresentation of defective viruses. Does this actually skew the data from clinical samples in which the IPDA will not be informative such as relative differences in proviral genomes? How much does this variation actually underestimate the "number" of cells harboring HIV?
Response 1: We appreciate Reviewer 3 for bringing up these important points.
We acknowledge that there could potentially be a higher proportion of negative samples in Indian cohorts relative to South African cohorts, given that there is a greater frequency of primer/probe sequence mismatches in Indian sequences against the IPDA-BC env forward primer. We also acknowledge that there may be an overrepresentation of defective viruses if our estimated drop in primer/probe binding success rate applies primarily to intact rather than defective viruses and vice versa. For example, if more intact viruses contain env regions with primer/probe mismatches in Indian samples relative to South African samples, but if this bias is absent in defective genomes, it could inflate the defective:intact genome ratio in Indian samples. To address this concern, large-scale full genome proviral sequencing of Indian samples is required; however, such a database does not currently exist.
We further acknowledge that sequence variation may lead to an underestimation of the number of cells harboring HIV in Indian relative to South African samples, given the higher frequency of primer/probe mismatches. However, the magnitude of underestimation is likely to vary across donors and geographic regions. For example, even within the same region, proviral sequence diversity is expected to be higher in individuals who experienced prolonged periods of viremia compared to those who initiated ART during acute infection (e.g., FRESH cohort). This is further complicated by the degree of nucleotide mismatches between the viral strain within each individual and the IPDA-BC primers/probes. In other words, the extent of underestimation is expected to be greater in individuals with more diverse proviral pools and lower sequence identity to the IPDA-BC primers/probes.
These considerations reflect the inherent nature of IPDA design: the presence of double-positive psi/env signals indicates a high likelihood of genome-intact viruses, whereas the absence of signal may result from either true absence, or it can be an indicator of primer/probe mismatches. In this context, the assay is best suited for longitudinal monitoring of changes in intact reservoir size within an individual, as reservoir diversity is expected to remain comparable across time points within the same donor on suppressive therapy. Accordingly, IPDA double-positive signals should be interpreted as a minimal estimate of intact proviral load.
We thank the reviewer for prompting this discussion and have now incorporated these considerations and the associated limitations in data availability in the Discussion section (lines 297–321).
Comment 2: Some of their findings would be potentially addressed if they performed IPDA utilizing samples from South Africa and India.
Response 2: We agree that the impact of sequence variation between subtype C HIV from South Africa and India on clinical samples should be further explored to quantify potential differences in IPDA-BC performance on Indian subtype C samples. In our study, we reported applying IPDA-BC to South African samples, and none of these samples failed the assay (Results, lines 254–256). While we do have access to samples from India, we currently do not have the resources to perform IPDA on this cohort. We fully agree that such data would be valuable and hope to address this in future studies.